# Influence of artificial intelligence in education on adolescents' social adaptability: The mediatory role of social support

Tinghong Lai[1,2], Chuyin Xie[1], Minhua Ruan[1], Zheng Wang[3], Hong Lu[1], Shimin Fu[1]*

1 Center for Brain and Cognitive Science, School of Education, Guangzhou University, Guangzhou, P.R. China, 2 Gannan Medical University, Guangzhou, P.R. China, 3 Management Center for Quality Education of Baiyun District, Guangzhou, P.R. China

* fusm@gzhu.edu.cn

**Data Availability Statement:** All relevant data are within the manuscript and its Supporting Information files.

**Funding:** This study received support from The Major Project of Guangzhou Educational Science

## Abstract

Artificial intelligence (AI) is widely used in the field of education at present, but people know little about its possible impacts, especially on the physical and mental development of the educated. It is important to explore the possible impacts of the application of artificial intelligence in education (AIEd) in order to avoid the possible adverse effects. Prior research has focused on theory to the exclusion of the psychological impact of AIEd, and the empirical research was relatively lacking. This study aimed to identify the influence of AIEd on adolescents' social adaptability via social support. A total of 1332 students were recruited using random sampling from 13 Artificial Intelligence Curriculum Reform Experimental Schools in Guangzhou, Southern China, completed the survey. There were 342 primary school students ($Mean_{age}$ = 10.6), 351 junior high school students ($Mean_{age}$ = 13.1), and 639 senior high school students ($Mean_{age}$ = 15.8). Results showed that AIEd has a negative impact on adolescents' social adaptability, and is significantly negatively correlated with social adaptability and family support, but there is no significant correlation with school support. AIEd could not only affect social adaptability directly, but also could affected it through the family support.

## Introduction

The application of artificial intelligence in education (AIEd) is a new trend in educational innovation and development. Particularly after the outbreak of COVID-19 in 2020, large-scale online teaching has become a big test of how artificial intelligence (AI) technology might enable education. Some researchers believe that AIEd brings more opportunities than threats [1, 2]. For example, Intelligent Tutoring System (ITS) has been found to be more effective than traditional teaching tools. An intelligent learning environment created based on a network tutoring system has been found to have a positive impact on the durability of education [3]. However, some researchers believe that there are potential risks of AIEd. For example, using intelligent technology to collect learners' data may cause safety and ethical problems due to data leakage [4]. Some researchers are even worried that AIEd may deviate from the purpose of education, and become a potential educational risk behavior due to the bias of its designers and executants.

Planning (No: 2020zd003). The funders had no role in study design, data collection and analysis, decision to publish, or preparation of the manuscrip.

**Competing interests:** The authors have declared that no competing interests exist.

Adolescents are the principal recipients of AIEd, and they are in a critical period in which they are very easily affected by the external environment [5, 6]. However, previous studies have only discussed the influence of AIEd on adolescent at the theoretical level, the lack of research into emotion and influencing factors research has always been a prominent problem in AIEd [7]. So, it is important to pay attention to the impact of AIEd on adolescents' physical and mental development.

## Application of artificial intelligence in education

The application of artificial intelligence in education (AIEd) can be understood as integrating artificial intelligence (AI) technology into the scenes of education. At present, a number of key AI technologies, including machine learning, knowledge mapping, and natural language processing are gradually being applied in education. In general, there are five typical ways in which AIEd is applied: an intelligent education environment, intelligent learning process support, intelligent educational evaluation, intelligent teacher assistance, and intelligent educational management and services [8]. In this study, AIEd refers to the universal application of AI technology in education, i.e., the new technologies used to improve teaching methods and enhance learning efficiency, expand teaching time-space environment, and improve teaching management and services. These can also be referred to as VR teaching, online learning, flat panel teaching, etc. Research to date has shown there are three main forms of AIEd being used in the Artificial Intelligence Curriculum Reform Experimental Schools in Guangzhou. One is the form of curriculum teaching, such as information technology courses, general technology courses, flat panel teaching, intelligent reading, etc. The second takes the form of interest classes, such as programming courses, assembling robots, etc. The third involves mass organization, such as 3D printing, Lego plug-ins, teaching boxes, etc.

## Application of artificial intelligence in education and social adaptability

Interpersonal relationships are important to one's social adaptability. Specifically, a good interpersonal relationship is conducive to social adaptability; otherwise, it is unfavorable [9]. According to the theory of social presence and the theory of social cue reduction which are based on cue filtering orientation, media communication is more prone than face-to-face communication to weaken the ability and expectation of individual to establish social interaction due to the lack of important nonverbal and situational cues, such as those involving vision, hearing and touch [10]. Although non-face-to-face online social contact produces less social pressure and lower social anxiety than real face-to-face social contact, most young people with social anxiety further escape from real social contact after obtaining social support through online [11], which is disadvantageous to their social adaptability.

Studies have shown that the application of artificial intelligence technology is conducive to individual development. For example, children who interact with robots show a high degree of creativity, promoting the development of their social ability [12], and wearable machines can enhance the expression ability of adolescents with autism spectrum disorders [13]. However, some studies have shown that the application of artificial intelligence technology is disadvantageous to individual development. For example, frequent use of intelligent electronic devices has a negative impact on adolescents' interpersonal relationships [14] and social adaptability [15], and the elderly who are cared for by robot partners feel more lonely and emotionally indifferent [16].

The application of artificial intelligence in education (AIEd) is based on computers and other media technologies, making it inseparable from the use of intelligent devices such as the Internet and electronic equipment. Education is a kind of social activity, and interaction and

cooperation are the core of the teaching process. However, AIEd makes machines become the intermediary connecting students and teachers, which changes the interpersonal relationship of teaching from human-human to human-machine-human. The changed space-time relationship of teaching leads to a decrease in real teacher-student interpersonal interaction, and the students' sense of social presence is weakened. AIEd has great situational difference from conventional teaching and lacks of sufficient nonverbal clues, situational clues and other important information, which is disadvantageous to adolescents' social adaptability.

## Social support, application of artificial intelligence in education and social adaptability

Social support refers to resources provided by others, and parents and peers are the most direct sources of social support for adolescents. Research has demonstrated that the more social support one receives, the better the social adaptability [17]. There is positive causality between social support and mental health [18], and a lack of social support is not conducive to adolescents' social adaptability.

Prior studies have indicated that parent-child relationships, parent-child communication, and peer relationships can affect adolescents' social adaptability [19, 20]. Adolescents who perceiving parents' support for basic psychological needs could predict their well-being, high self-esteem, and sense of choice about their own behavior [21]. According to the Parental Acceptance-Rejection Theory, when an individual is rejected by his/her parents, his/her emotional connection with supportive caregivers can buffer or compensate for the negative impact of parental rejection [22]. As peers are the most "important others" of adolescents except for their parents, peer acceptance could significantly predict the interpersonal adaptability of primary school students, and a good peer relationship can excellently predict emotional expression ability and social adaptability in adolescents [23], while poor peer relationships may have a negative impact on their social adaptability [24]. In addition, prior studies have also suggested that a good teacher-student relationship can promote the school adaptability and social adaptability of adolescents, and has an impact on parent-child relationships and peer relationships [25–28].

82% of information in teaching is transmitted through nonverbal communication. Nonverbal intimate behaviors, such as facing students, smiling, approaching students, eye contact and communication, voice cadence, and positive posture [29] are the center of effective teaching. The more nonverbal intimate behaviors, the better the effect on students' emotional learning [30]. The application of artificial intelligence in education (AIEd) reduces the nonverbal intimacy behaviors between teachers and students, and thus their sense of social presence and interpersonal interaction is weakened. In addition, individuals' over-reliance on intelligent devices may lead to a reduction in parent-child communication, peer communication and teacher-student interaction. AIEd may not only change the way of communication but also the relationships with teachers, peers, and parents. To sum up, we propose hypothesis 1: AIEd affects adolescents' social adaptability, and is related to perceived social support. Hypothesis 2: Perceived social support plays intermediary role between AIEd and social adaptability. The specific hypothetical model is shown in Fig 1.

## Materials and methods

### Participants and method

1332 students recruited through random sampling from a total of 28 classes across 13 schools participated in the survey. All schools were Artificial Intelligence Curriculum Reform

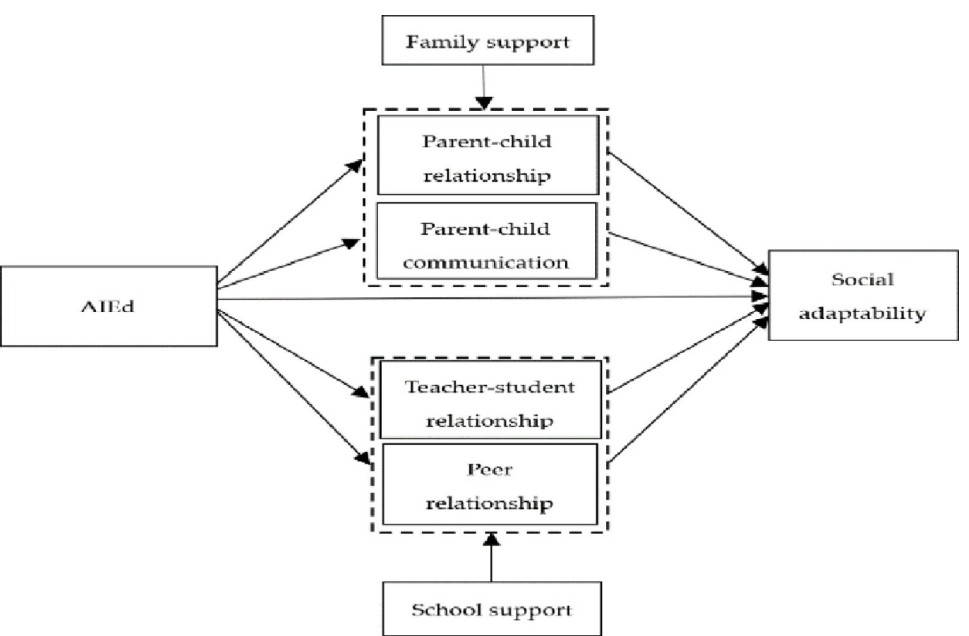

**Fig 1. Model of mediatory role of social support between AIEd and social adaptability.**

Experimental Schools in Guangzhou, and participants included 342 primary school students ($Mean_{age}$ = 10.6), 351 junior high school students ($Mean_{age}$ = 13.1), and 639 senior high school students ($Mean_{age}$ = 15.8). Students who met the following criteria were not eligible for the study: unable to understand the terms in the questionnaire or leaving more than 30% of the items uncompleted. Missing values of the included data were replaced with averages. Before completing the survey, all participants gave written informed consent. This study was approved by the Ethics Committee of Guangzhou University (No: GZHU2020010). Questionnaires were completed anonymously within 40 minutes and collected on the spot. A total of 1318 valid samples were obtained ($Mean_{age}$ = 13.56), giving an effective return rate of 98.95%. SPSS25.0 and model 4 of Process of SPSS as compiled by Hayes (2013) were used for statistical analysis.

### Measures tools

Artificial Intelligence Usage Questionnaire. The Mobile Phone Usage Questionnaire (Wang & Zhang, 2018) was used to investigate the usage of AI among adolescents. One of the items, "*Are you using AI or smart phones, tablets and other intelligent devices to learning*", is scored with options 1 = *yes* and 2 = *no*. Based on their response, students were divided into the artificial intelligence group and non- artificial intelligence group.

Parent-Child Relationship Scale was adopted to measure the quality of parent-child relationships (Cronbach's alpha = 0.92). It comprises 18 items, scored on a 5-point scale from 1 = "*completely unqualified* to 5 = "*very qualified*". The higher the total score, the better the parent-child relationship.

Parent-Child Communication Scale (Wang, et al.,2006) was adopted to measure the communication between parents and children (Cronbach's alpha = 0.89). It comprises 20 items (10 of which are reverse scored), scored on a 5-point scale from 1 = "*strongly disagree*" to 5 = "*strongly agree*". The higher the total score, the better the communication between parents and children.

Teacher-Student Relationship Scale (Chen & Li, 2009) was adopted to measure the teacher-student relationship (Cronbach's alpha = 0.91). The scale comprises 7 items scored on a 5-point scale from 1 = "*completely inconsistent*" to 5 = "*very consistent*", with item 7 reverse scored. The higher the total score, the better the relationship between teachers and the student.

Peer-Relationship Scale (Chen & Zhu, 1997) was adopted to measure peer relationships (Cronbach's alpha = 0.83). It comprises 18 items scored on a 6-point scale from 1 = "*completely inconsisten*t" to 6 = "*completely consistent*". Nine items (items 1, 2, 5, 6, 9, 10, 13, 14, 17) are reversed scored, The higher the total score, the better the relationships with peers.

Social Adaptability Scale (Zheng, 1999) was adopted to measure social adaptability (Cronbach's alpha = 0.80). It comprises 20 items, and uses a 3-point scoring method (1 = "*Yes*", 2 = "*Uncertain*", 3 = "*No*"). The higher the total score, the better the social adaptability.

## Results

### Common method deviation test

Because all the variables under investigation were measured by scales, the Harman Single Factor Test was used to look for possible common method deviation. Results showed that there were 23 factors with characteristic values greater than 1 and the first factor explained the variation of 20.12%, which is less than the critical standard of 40%. Therefore, there is no serious problem of Common Method Deviation.

### Difference in social adaptability between artificial intelligence group and non- artificial intelligence group

One way ANOVA was used to analyze the difference in social adaptability between the artificial intelligence group (AI group) and non-artificial intelligence group (non-AI group). Results show that there was a significant social adaptability difference between them [$F (1, 1318) =$ 10.068, $p < 0.01$], indicating that AIEd has an impact on adolescents' social adaptability. (See Table 1).

### Correlation analysis of application of artificial intelligence in education, social support, and social adaptability

Results of Bivariate Correlation Analysis showed that social adaptability was significantly negatively correlated with the application of artificial intelligence in education (AIEd) ($r = -0.087$, $p < 0.01$), significantly positively correlated with parent-child relationship, parent-child communication, and teacher-student relationship ($r = 0.307$, $p < 0.01$; $r = 0.405$, $p < 0.01$; $r = 0.166$, $p < 0.01$), but not significantly correlated with peer relationship ($r = -.007$, $p > 0.05$). AIEd was significantly negatively correlated with parent-child relationship and parent-child communication ($r = - 0.054$, $p < 0.05$; $r = -0.086$, $p < 0.05$), but not significantly correlated

**Table 1. Differences in social adaptability of students between AI group and non-AI group.**

| Group | N | M | SD | t | p |
|---|---|---|---|---|---|
| AI group | 998 | 3.10 | 13.96 | - | - |
| Non-AI group | 320 | 6.04 | 14.59 | - | - |
| | | | | 3.173 | 0.002** |

*Note:*[*] = $p < 0.05$

[**] = $p < 0.01$

[***] = $p < 0.001$.

**Table 2. Correlation analysis results of AIEd, social support and social adaptability.**

| Variables | M (SD) | 1 | 2 | 3 | 4 | 5 |
|---|---|---|---|---|---|---|
| 1. AIEd | 1.24 (0.429) | - | - | - | - | - |
| 2. P-C relationship | 63.11 (14.447) | -.054* | - | - | - | - |
| 3. P-C communication | 131.52 (27.839) | -.086* | .734** | - | - | - |
| 4. T-S relationship | 26.66 (4.769) | -.048 | .310** | .288** | - | - |
| 5.Peer relationship | 54.76 (8.793) | .023 | .174** | .153** | .147** | - |
| 6. Social Adaptability | 5.34 (14.468) | -.087** | .307** | .405** | .166** | -.007 |

Note:* = p < 0.05

** = p < 0.01

*** = p < 0.001. P-C = Parent-child, T-S = Teacher-student.

with teacher-student relationship or peer relationship (r = -0.048, $p > 0.05$; r = 0.023, $p > 0.05$). That is to say, AIEd is significantly negatively correlated with family support, but not significantly correlated with school support. Family support could predict social adaptability positively, which is consistent with previous research results. AIEd is significantly negatively correlated with social adaptability and family support significantly. (See Table 2).

## Examination of the intermediary effect of social support between application of artificial intelligence in education and social adaptability

According to the results of correlation analysis, the application of artificial intelligence in education (AIEd) is significantly related to social adaptability and family support, which can be analyzed further. First, AIEd was taken as the independent variable, with social adaptability, parent-child relationship, and parent-child communication as dependent variables for regression analysis. Next, parent-child relationship and parent-child communication were taken as independent variables, with social adaptability as the dependent variable for regression analysis. Finally, social adaptability was taken as the dependent variable, AIEd as the independent variable, and parent-child relationship and parent-child communication as intermediary variables for intermediary effect analysis.

As shown in Table 3, AIEd negatively predicts social adaptability, parent-child relationship and parent-child communication ($\beta$ = -.087, $p < 0.01$; $\beta$ = -.054, $p < 0.05$; $\beta$ = -.086, $p < 0.01$), while parent-child relationship and parent-child communication positively predict social adaptability ($\beta$ = .307, $p < 0.001$; $\beta$ = .405, $p < 0.001$). Furthermore, model 4 of Process for SPSS as compiled by Hayes (2013) was used to test the effect size and significance, in which the bootstrap was 5000 times. As shown in Table 4, the total effect of AIEd on social adaptability was [-.203] and the confidence interval did not include 0 [-.327, -.080], indicating that the total effect is significant. The direct effects of AIEd on social adaptability were [-.121] and [-.165], with 95% confidence intervals of [-.238, -.003] and [-.283, -.046] respectively; the confidence intervals do not include 0, indicating that the direct effect is significant. The intermediary effect of parent-child relationship and parent-child communication between AIEd and social adaptability were [-.039] and [-.083], with 95% confidence intervals of [-.081, -.001] and [-.136, -.035] respectively; the confidence intervals do not include 0, indicating that the intermediary effect is significant. We tested the mediatory model of parent-child relationship between AIEd and social adaptability, and the parent-child communication between AIEd and social adaptability, the mediatory moedel showed an excellent fit to the data: $\chi^2$/df = 4.81, comparative fit index (CFI) = 1.00, root mean square error of approximation (RMSEA) = 0.019;

**Table 3. Effect analysis of AIEd, social support and social adaptability.**

| Variables | Social Adaptability | | P-C relationship | | P-C communication | |
|---|---|---|---|---|---|---|
| | $\beta$ | $t$ | $\beta$ | $t$ | $\beta$ | $t$ |
| AIEd | -.087 | −3.173** | -.054 | −1.976* | -.086 | −3.132** |
| $R^2$ | .007 | | .002 | | .007 | |
| $F$ | 10.068 | | 3.903 | | 9.808 | |
| **Variables** | **Social Adaptability** | | **P-C relationship** | | **P-C communication** | |
| | $\beta$ | $t$ | $\beta$ | $t$ | $\beta$ | $t$ |
| P-C relationship | .307 | 11.755*** | - | - | - | - |
| $R^2$ | .094 | | - | | - | |
| $F$ | 138.187 | | - | | - | |
| **Variables** | **Social Adaptability** | | **P-C relationship** | | **P-C communication** | |
| | $\beta$ | $t$ | $\beta$ | $t$ | $\beta$ | $t$ |
| P-C communication | .405 | 16.150*** | - | - | - | - |
| $R^2$ | .164 | | - | | - | |
| $F$ | 260.818 | | - | | - | |

*Note:* * = $p < 0.05$

** = $p < 0.01$

*** = $p < 0.001$. *P-C = Parent-child.*

$\chi^2$/df = 3.92, comparative fit index (CFI) = 1.00, root mean square error of approximation (RMSEA) = 0.023. (See Fig 2).

## General discussion

From the results above, we conclude that the application of artificial intelligence in education (AIEd) negatively affects adolescent's social adaptability, and that family support (a form of social support) plays an intermediary role between AIEd and social adaptability. This finding is consistent with our hypothesis. Prior studies have demonstrated that artificial intelligence (AI) technology is conducive to individual development, but most of them take adolescents with weak social functions (such as, the autistic or hearing-impaired) as objects [13]. Different objects may lead to different results; this study takes normal adolescents as objects, and the results show that AIEd has a negative impact on social adaptability. Certainly, AIEd has positive effects on adolescents, such as promoting learning, but it cannot completely replace the functions of people. Otherwise, it will do more harm than good.

The parent-child relationship is an important social relationship and indeed the first that individuals experience. Bowlby (1973) believed that one's early parent-child relationships

**Table 4. Intermediary effect test of social support (family support).**

| Effect type | Effect value | SE | Relative effect quantity | Bootstrap (95%CI) |
|---|---|---|---|---|
| Total effect | -.203 | .063 | 100% | [-.327, -.080] |
| Direct effect | -.121 | .060 | 59.24% | [-.238, -.003] |
| Intermediary effect of P-C communication | -.083 | .026 | 40.80% | [-.136, -.035] |
| Total effect | -.203 | .063 | 100% | [-.327, -.080] |
| Direct effect | -.165 | .060 | 80.87% | [-.283, -.046] |
| Intermediary effect of P-C relationship | -.039 | .020 | 19.17% | [-.081, -.001] |

Note: P-C = Parent-child.

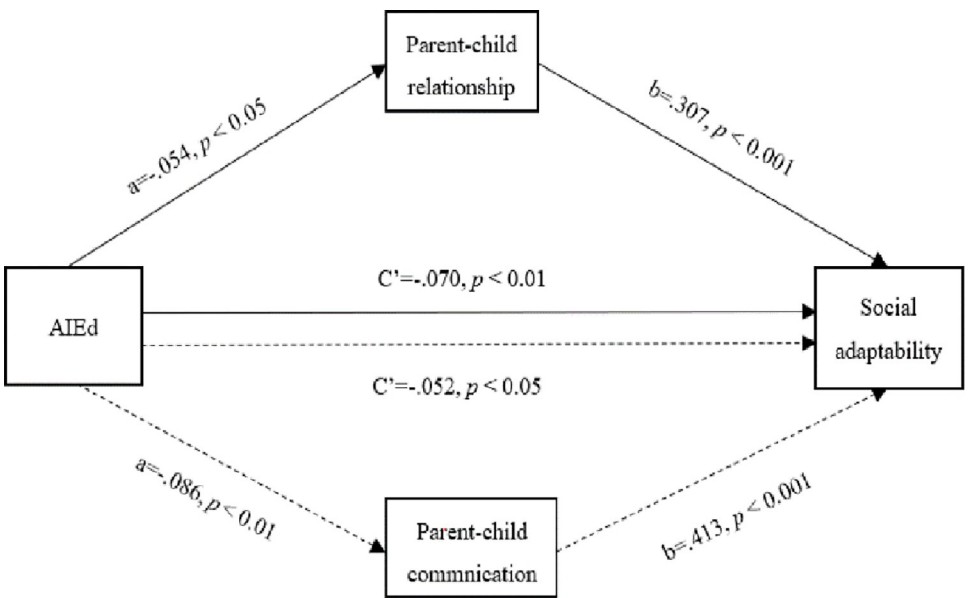

**Fig 2. Mediatory model of family support system between AIEd and social adaptability.**

affect the construction of a safe internal work model, which will later affect teacher-student and peer-relationships, thus affecting social adaptability. At present, AIEd is in the initial stage in China. Students are not deeply involved, and their main social relationships are family relationships, so their social adaptability is more obviously affected by family support than school support.

However, this study's findings must be understood within the context of its specific limitations. The cross-sectional approach used to collect data makes it hard to establish a causal relationship between AIEd and social adaptability. Moreover, social adaptability is a dynamic process, and the impact of AIEd may present phased characteristics. Therefore, longitudinal data should be collected. Furthermore, this study principally relied on participants' self-reports. Further research should utilize multiple data collection methods, such as parental report and peer report, or empirical research, which may help assess their relationship more accurately and reduce the common method variance. Finally, we distinguished the AI group and non-AI group based on participants' self-report; a more rigorous grouping method could be adopted in future research.

## Conclusion

This study aimed to identify the impact of the application of artificial intelligence in education (AIEd) on adolescents' social adaptability from the perspective of social support. Results show that AIEd negatively affects adolescents' social adaptability. Family support, a form of social support, plays an intermediary role between AIEd and adolescents' social adaptability. Future research should further explore the impact of AIEd on individual physical and mental development to determine possible risks.

## Supporting information

**S1 File. Social adaptability and social support- minimized data.**
(SAV)

## Acknowledgments

We would like to acknowledge all participants for their time, effort, and contribution.

## Author Contributions

**Conceptualization:** Hong Lu.

**Data curation:** Tinghong Lai, Chuyin Xie.

**Funding acquisition:** Shimin Fu.

**Investigation:** Chuyin Xie, Minhua Ruan.

**Resources:** Zheng Wang.

**Writing – original draft:** Tinghong Lai.

**Writing – review & editing:** Tinghong Lai, Shimin Fu.

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
