## [Decision Letter · Decision Letter 0]

7 Dec 2022

PONE-D-22-21872Influence of Artificial Intelligence in Education on Adolescent’s Social AdaptabilityPLOS ONE

Dear Dr. Lai,

Thank you for submitting your manuscript to PLOS ONE. After careful consideration, we feel that it has merit but does not fully meet PLOS ONE’s publication criteria as it currently stands. Therefore, we invite you to submit a revised version of the manuscript that addresses the points raised during the review process.

We look forward to receiving your revised manuscript.

Kind regards,

Md Zia Uddin

Academic Editor

PLOS ONE

Journal Requirements:

2. During your revisions, please note that a simple title correction is required: "Adolescent's" should be replaced by "Adolescents' ". Please ensure this is updated in the manuscript file and the online submission information.

 “This study was supported by The Major Project of Guangzhou Educational Science Planning:” Research on the influence of AI on the social adaptability of educational objects” (Project No: 2020zd003)”

5. Please amend either the title on the online submission form (via Edit Submission) or the title in the manuscript so that they are identical

6. PLOS requires an ORCID iD for the corresponding author in Editorial Manager on papers submitted after December 6th, 2016. Please ensure that you have an ORCID iD and that it is validated in Editorial Manager. To do this, go to ‘Update my Information’ (in the upper left-hand corner of the main menu), and click on the Fetch/Validate link next to the ORCID field. This will take you to the ORCID site and allow you to create a new iD or authenticate a pre-existing iD in Editorial Manager. Please see the following video for instructions on linking an ORCID iD to your Editorial Manager account: https://www.youtube.com/watch?v=_xcclfuvtxQ.

7. We note that you have indicated that data from this study are available upon request. PLOS only allows data to be available upon request if there are legal or ethical restrictions on sharing data publicly. For more information on unacceptable data access restrictions, please see http://journals.plos.org/plosone/s/data-availability#loc-unacceptable-data-access-restrictions.

Reviewers' comments:

Reviewer's Responses to Questions

**Comments to the Author**

1. Is the manuscript technically sound, and do the data support the conclusions?

Reviewer #1: Yes

Reviewer #2: Yes

2. Has the statistical analysis been performed appropriately and rigorously? 

Reviewer #1: Yes

Reviewer #2: Yes

3. Have the authors made all data underlying the findings in their manuscript fully available?

Reviewer #1: Yes

Reviewer #2: Yes

4. Is the manuscript presented in an intelligible fashion and written in standard English?

Reviewer #1: Yes

Reviewer #2: Yes

5. Review Comments to the Author

Reviewer #1: This is a very well designed and well written paper.

Analysis is sophistocated enough.

The results are new.

Here are my suggestions:

1 - Only main effects and mediations are mentioned. Please also test the effects by social sttaus and gender/sex.

2- Please add data on fit statistics.

3- Please report standardized coefficients (rather than unstandardized b) in the paper and also in figures.

4- What is data on ethnicity / social class?

After these minor changes, I am happy to recoimmend acceptance.

Reviewer #2: In this paper, authors analyze the influence of AI in education on adolescent’s social adaptability. The paper is well structured and well-written. I have few comments:

In Abstract: Please be consistent, authors used AIed and AIEd variations. Moreover, please use full words instead of abbreviations for the very first time.

While applying ANOVA, how the the problem was formulated for non-AI group? Does AI usage questionnaires represent the AI group and rest parent-child relationship scholar or teacher student scales lies in non-AI group?

Did authors try other methods then ANOVA? If not, then why ANOVA was chosen, please elaborate.

6. PLOS authors have the option to publish the peer review history of their article (what does this mean?). If published, this will include your full peer review and any attached files.

Reviewer #1: No

Reviewer #2: No

---

## [Author Response · Author response to Decision Letter 0]

15 Dec 2022

Responds to reviewer#1

1 - Only main effects and mediations are mentioned. Please also test the effects by social sttaus and gender/sex.

Respond 1: In this study, we mainly consider to explore the impact of AIEd on adolescents’ social adaptability, and the mediatory role of social support between AIEd and social adaptability. Of course, gender, social status, race, and all, that you mentioned, may be influencing factors, but we have not included them in this study. We will consider the role of these factors in our future research. Thank you for your review and suggestions. 

2- Ple ase add data on fit statistics.

Respond 2：we add data on fit statistics in the revision manuscript as follow:

We tested the mediatory model of parent-chlid relationship between AIEd and social adaptability, and the parent-child communication between AIEd and social adaptability, the mediatory moedel showed an excellent fit to the data: �2/df = 4.81, comparative fit index (CFI) = 1.00, root mean square error of approximation (RMSEA)= 0.019; �2/df = 3.92, comparative fit index (CFI) =1.00, root mean square error of approximation (RMSEA)= 0.023.

3- Please report standardized coefficients (rather than unstandardized b) in the paper and also in figures.

Respond 3: In our manusript, β (Beta) in the paper is the standardized coefficients rather than understandardized coefficients B. and we re-report the standardizd coefficients in the figures.

4- What is data on ethnicity / social class?

Respond 4: Thank you for your review and suggestion. In this study, we did not consider the impact of ethnicity/social class, so we did not collect those data. In our future study, we will collect the ethnicity/social class data, and consider their impact.

Responds to reviewer#2

In Abstract: Please be consistent, authors used AIed and AIEd variations. Moreover, please use full words instead of abbreviations for the very first time.

Respond: We are sorry for the misspelling. We have replaced AIed with AIEd, and they are identical now. Besides, as your suggestion, we have used full words instead of abbreviations for the very first time.

While applying ANOVA, how the the problem was formulated for non-AI group? Does AI usage questionnaires represent the AI group and rest parent-child relationship scholar or teacher student scales lies in non-AI group?

Respond: Although subjects in our study are students come from Artificial Intelligence Curriculum Reform Experimental Schools in Guangzhou, but not all students use artificial intelligence for learning. So, the AI usege questionnaires we used in study has one items: “Are you using AI or smart phones, tablets and other intelligent devices to learning”, is scored with options 1= yes and 2 = no. Based on their response, students were divided into the artificial intelligence group and non- artificial intelligence group, 1 is AI group and 2 is non-AI group. The parent-child relationship scale and the teacher-student relationship scale are filled in by all students participating in the survey, whether they use artificial intelligence for learning or not. All subjects completed all questionnaires.

Did authors try other methods then ANOVA? If not, then why ANOVA was chosen, please elaborate.

Respond: We also carried out average value analysis, and the result showed that they were consistent with the result in manuscript, it is significant. Because ANOVA can determine the influence of controllable factors on the research results, so we choose present the result of ANOVA in manuscript.

---

## [Decision Letter · Decision Letter 1]

3 Mar 2023

Influence of Artificial Intelligence in Education on Adolescents' Social Adaptability: The Mediatory Role of Social Support

PONE-D-22-21872R1

Dear Dr. Lai,

We’re pleased to inform you that your manuscript has been judged scientifically suitable for publication and will be formally accepted for publication once it meets all outstanding technical requirements.

Kind regards,

Md Zia Uddin

Academic Editor

PLOS ONE

Additional Editor Comments (optional):

Reviewers' comments:

Reviewer's Responses to Questions

**Comments to the Author**

1. If the authors have adequately addressed your comments raised in a previous round of review and you feel that this manuscript is now acceptable for publication, you may indicate that here to bypass the “Comments to the Author” section, enter your conflict of interest statement in the “Confidential to Editor” section, and submit your "Accept" recommendation.

Reviewer #1: All comments have been addressed

Reviewer #2: All comments have been addressed

2. Is the manuscript technically sound, and do the data support the conclusions?

Reviewer #1: Yes

Reviewer #2: Yes

3. Has the statistical analysis been performed appropriately and rigorously? 

Reviewer #1: Yes

Reviewer #2: Yes

4. Have the authors made all data underlying the findings in their manuscript fully available?

Reviewer #1: Yes

Reviewer #2: Yes

5. Is the manuscript presented in an intelligible fashion and written in standard English?

Reviewer #1: Yes

Reviewer #2: Yes

6. Review Comments to the Author

Reviewer #1: Congratulations. The paper can be accepted because the revision is satisfactory. The analysis is added, statistics are appropriate, and figures and tables are informative.

Reviewer #2: (No Response)

7. PLOS authors have the option to publish the peer review history of their article (what does this mean?). If published, this will include your full peer review and any attached files.

Reviewer #1: No

Reviewer #2: **Yes: **Farzan Majeed Noori

---

## [Editor Report · Acceptance letter]

9 Mar 2023

PONE-D-22-21872R1 

Influence of Artificial Intelligence in Education on Adolescents' Social Adaptability: The Mediatory Role of Social Support 

Dear Dr. Lai:

I'm pleased to inform you that your manuscript has been deemed suitable for publication in PLOS ONE. Congratulations! Your manuscript is now with our production department. 

Kind regards, 

on behalf of

Dr. Md Zia Uddin 

Academic Editor

PLOS ONE